Methods

# Novel imaging tools to study mitochondrial morphology in *Caenorhabditis elegans*

Miriam Valera-Alberni, Pallas Yao, Silvia Romero-Sanz, Anne Lanjuin, William B Mair

**Mitochondria exhibit a close interplay between their structure and function. Understanding this intricate relationship requires advanced imaging techniques that can capture the dynamic nature of mitochondria and their impact on cellular processes. However, much of the work on mitochondrial dynamics has been performed in single celled organisms or in vitro cell culture. Here, we introduce novel genetic tools for live imaging of mitochondrial morphology in the nematode *Caenorhabditis elegans*, addressing a pressing need for advanced techniques in studying organelle dynamics within live intact multicellular organisms. Through a comprehensive analysis, we directly compare our tools with existing methods, demonstrating their advantages for visualizing mitochondrial morphology and contrasting their impact on organismal physiology. We reveal limitations of conventional techniques, whereas showcasing the utility and versatility of our approaches, including endogenous CRISPR tags and ectopic labeling. By providing a guide for selecting the most suitable tools based on experimental goals, our work advances mitochondrial research in *C. elegans* and enhances the strategic integration of diverse imaging modalities for a holistic understanding of organelle dynamics in living organisms.**

## Introduction

Mitochondria, essential organelles in eukaryotic cells, exhibit a dynamic and interconnected network architecture crucial for their functional adaptability. This intricate network is governed by coordinated processes such as membrane fusion, fission, and mitochondrial trafficking, collectively referred to as mitochondrial dynamics. The ability of mitochondria to undergo morphological changes is pivotal for their functional capacity to respond to nutrient availability, molecular signals, and propagate mitochondrial genomes (Liesa & Shirihai, 2013). Disruptions in this network organization have been linked to mitochondrial dysfunction, thereby contributing to the onset of various pathophysiological conditions, including neurodegenerative diseases, cardiovascular disorders,

cancer, or metabolic diseases (Chen et al, 2023). Consequently, research into mitochondrial dynamics has become integral to comprehending mitochondrial function in disease states. Despite this growing awareness, a significant gap remains in the availability of optimal imaging tools to visualize mitochondrial morphology in multicellular living organisms. Much of the work to date has been performed in single-celled organisms such as *Saccharomyces cerevisiae* or using in vitro cell culture. Although these systems have many advantages for understanding the molecular mechanisms governing mitochondrial regulation, they are more limited in their utility when it comes to physiology. Conversely, where mouse models support examination of how changes to mitochondrial dynamics impact physiology and pathophysiology, in vivo imaging is a challenge. This study aims to fill this gap by introducing novel imaging tools tailored for the nuanced visualization of mitochondria in the nematode worm *Caenorhabditis elegans*, offering new opportunities for the study of mitochondrial dynamics in disease conditions and longitudinally throughout an animal's life.

Powerful genetic approaches have expedited the molecular characterization of multiple mitochondrial processes, making *C. elegans* an ideal animal model for understanding the organismal roles of mitochondria (Onraet & Zuryn, 2024). Many fundamental genetic, biochemical, and structural features of mitochondria remain conserved between mammals and *C. elegans* (Ha et al, 2022; Campbell & Zuryn, 2024), including key components of the mitochondrial dynamics machinery and receptor complexes of the outer (OMM) and inner (IMM) mitochondrial membranes. In line with this, *C. elegans* provides a particular useful platform for visualizing mitochondrial morphology in vivo. As compared with single cell models such as yeast and cultured cells, *C. elegans* offers a complex organization of differentiated tissues, including a defined nervous system, hypodermis, gastrointestinal system, muscle, and germ lineage in which to study cell-type specific and cell non-autonomous features of mitochondrial function (Burkewitz et al, 2015; Onraet & Zuryn, 2024). Furthermore, the transparency of its cuticle allows for noninvasive, high-resolution imaging of mitochondria in live animals. This is in contrast with studies in mice, where changes in mitochondrial morphology are derived from biopsied tissues extracted from euthanized animals. Imaging in *C. elegans* can be conducted without tissue fixation, thus allowing

Department of Molecular Metabolism, Harvard TH Chan School of Public Health, Boston, MA, USA

Correspondence: wmair@hsph.harvard.edu

the visualization of real-time mitochondrial dynamics under distinct physiological conditions, which facilitates longitudinal studies. Indeed, the short lifespan of *C. elegans* enables researchers to conduct aging experiments that otherwise would require substantially more time in mice (mean lifespan of 2 yr) (Apfeld & Alper, 2018). Finally, the genetic tractability of worms allows targeted manipulation of mitochondrial components, enabling precise investigations into the molecular mechanisms governing mitochondrial dynamics.

The potential of *C. elegans* for cell biology in vivo, however, has been hindered by a lack of development of tools to visualize organelles for quantitative microscopy. We aimed to address this gap by engineering a suite of novel transgenic stains to study mitochondrial dynamics in a living animal. With the advent of CRISPR knock in gene editing, sophisticated approaches to generate new tools and visualize endogenous proteins are now feasible. We demonstrate that these new tools overcome some of the caveats posed by previously used methods to visualize mitochondria in *C. elegans*. Ultimately, we provide a comparison of past and current tools, highlighting specific caveats and giving researchers a data led approach to determine best imaging practices. Together, we believe that the strains presented here provide the foundation for in vivo cell biology studies of mitochondria in *C. elegans*.

## Results

### Limitations of current tools to study mitochondrial structure in *C. elegans*

Increasing attention has been directed towards exploring the vital role of mitochondrial dynamics in influencing mitochondrial function and, consequently, its impact on organismal physiology. Advanced imaging tools are essential for the analysis of mitochondrial dynamics, yet the instruments used to investigate mitochondrial morphology in *C. elegans* remain outdated. The primary indicator of mitochondrial morphology in most publications involving *C. elegans* was established in 2006, a mitochondrial-targeted GFP (GFP$^{mit}$) under the control of the muscle-specific *myo-3p* promoter (Benedetti et al, 2006). However, this tool presents several caveats that hinder its utility in cell biology studies. First, this model was generated using extrachromosomal plasmid arrays and integrated in the genome via gamma irradiation in unknown copy number and at a random locus (Benedetti et al, 2006). Variability in baseline expression levels make quantitative microscopy nearly impossible within the same strain, and especially across multiple conditions. In addition, possibly due to off-target effects of gamma irradiation or the high expression levels of the multicopy array, the mito-GFP strain displays slight developmental delay and reduced brood size (Fig 1A). Second, as a mito-targeted fluorophore that is nonintegrated into the membrane or constitutively residing in the lumen, visualization of mitochondrial networks relies on the fidelity of mitochondrial protein import. Mitochondrial protein import of GFP$^{mt}$ is inconsistent even in young day 1 worms, as shown by inadequate GFP labeling of the mitochondrial network and cytosolic signal in some animals (Fig 1B). In addition, lifespan analysis reveals that the mito-GFP strain is long

lived (Fig 1C) suggesting broader effects on physiology and mitochondrial function. Despite the fact that extrachromosomal arrays offer a reliable and fast means to express transgenes in *C. elegans*, they also have limitations. Since extrachromosomal arrays are semistable, only a fraction of the animals in a transgenic extrachromosomal array line express and propagate the transgene (Praitis et al, 2001). This leads to the laborious task of selecting the animals with the integrated transgene before starting any experimental approach. In addition, variations in array copy number and repeat sequence silencing can lead to heterogeneity in expression level and pattern of the transgene between individuals. Many existing tools in *C. elegans*, including those made by us previously, use extrachromosomal TOMM-20$^{aa1-49}$::GFP arrays that selectively visualize mitochondria within muscle, facilitated by a muscle-specific promoter (Fig S1A). However, this does not permit imaging of mitochondria in other tissues (Fig S1A). In addition to the labor-intensive process of individually selecting each GFP-positive worm, these animals exhibit minor developmental delay and reduced brood size (Fig S1B), as well as slight extension in lifespan (Fig S1C).

Inconsistency in mitochondrial labelling is also true for mitochondrial dyes, such as tetramethylrhodamine ethyl ester (TMRE), which are impacted by mitochondrial membrane potential (Chazotte, 2009; Gökerküçük et al, 2020). Depolarized mitochondria, which tend to appear in disease-prone conditions such as aging (Berry et al, 2023), display decreased membrane potential and fail to sequester TMRE (Fig S1D). Therefore, staining with TMRE should only be used to monitor differences in membrane potential, rather than for assessing mitochondrial network morphology. Moreover, staining by TMRE varies across tissues due to its infiltration, proving highly effective in the hypodermis or the intestine but showing lower infiltration in muscle and head tissues (Fig 2). Thus, mitochondrial labeling with dyes, such as TMRE, is not well suited for continual studies as their signal is dependent on mitochondrial properties that change with time and location.

To circumnavigate these caveats, we generated several new fluorescence reporters that are expressed somatically or ubiquitously to visualize mitochondrial morphology in live *C. elegans* via two strategies: first, via CRISPR-assisted single copy knock in of somatically expressed TOMM-20$^{aa1-49}$::GFP/mCherry and second, via CRISPR in frame integration of fluorophores into the loci of genes encoding endogenous mitochondrial membrane proteins.

### Visualizing mitochondria across multiple tissues in *C. elegans*

Advancements in precision gene-editing techniques, particularly in the context of manipulating genes within *C. elegans*, have yielded improved tools to probe mitochondria. We generated novel strains by tagging components of the OMM to fluorescence markers such as GFP, mCherry, or mScarlet. First, we generated two novel strains carrying the 1–49aa transmembrane domain fragment of the OMM translocase TOMM-20 (TOMM-20) fused to GFP or mCherry, driven by the *eft-3* promoter for ubiquitous tissue expression (TOMM-20$^{aa1-49}$::GFP/mCherry) (Fig S2A). Using CRISPR/Cas9 editing via the SKI LODGE method (Silva-García et al, 2019), this construct was integrated in single copy at a defined intergenic region on Chromosome V, previously characterized to give stable expression with

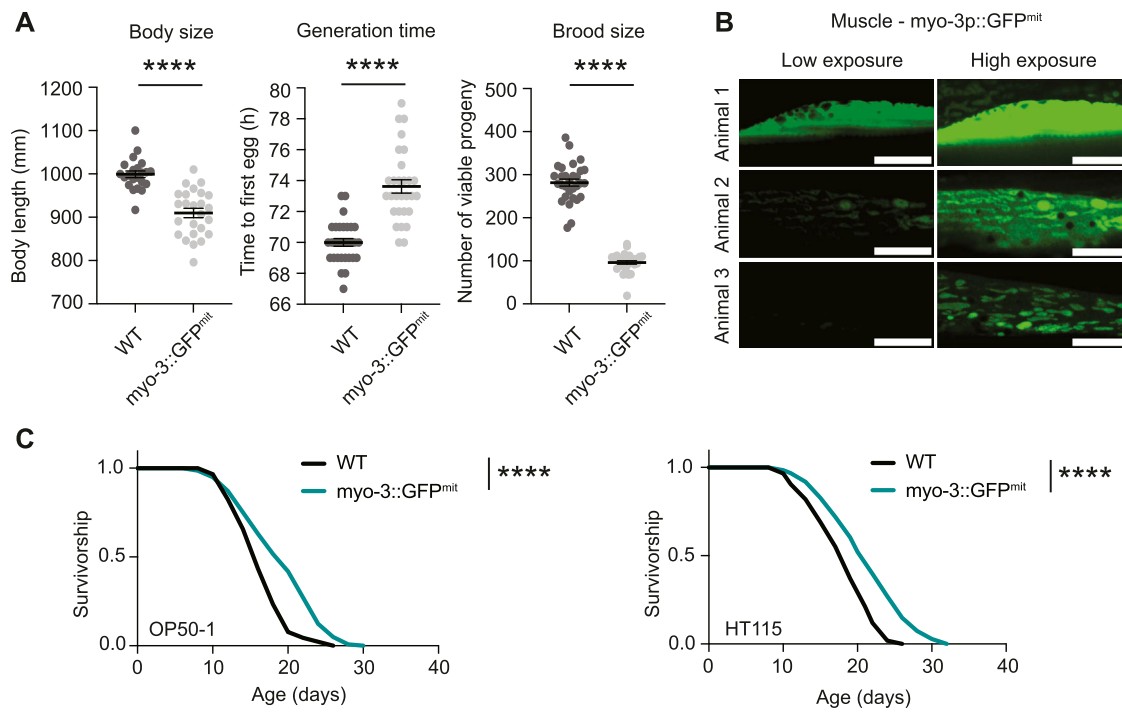

**Figure 1. Limitations of current mitochondrial imaging tools in *C. elegans*.**
**(A)** Healthspan analysis of the myo-3::GFP^mit strain compared with WT N2 worms, showing body size (left graph), generation time (middle), and brood size (right). Body size measurements were performed with 20–25 individual worms. Generation time and brood size measurements were performed with 30 worms pooled from two independent experimental repeats. All values are presented as mean ± SEM. ****$P < 0.0001$ versus the respective WT group (nonparametric Kruskal–Wallis's test). **(B)** Representative fluorescence pictures depicting mitochondrial networks in the body wall muscle of three different worms on day 2 of adulthood. The upper panel displays a muscle cell with a predominantly cytosolic GFP signal from myo-3p::GFP^mit, whereas the middle and bottom panels show a gradual labeling of mitochondria by GFP. For comparison, the left images are in low exposure, whereas the right images are in high exposure. Scale bar: 10 $\mu$m. **(C)** Survival curve of the myo-3::GFP^mit strain compared with WT in OP50-1 (left lifespan) and HT115 (right lifespan) bacteria. ****$P < 0.0001$ indicating significant difference compared with WT animals by one-way ANOVA with Tukey's multiple comparisons test.

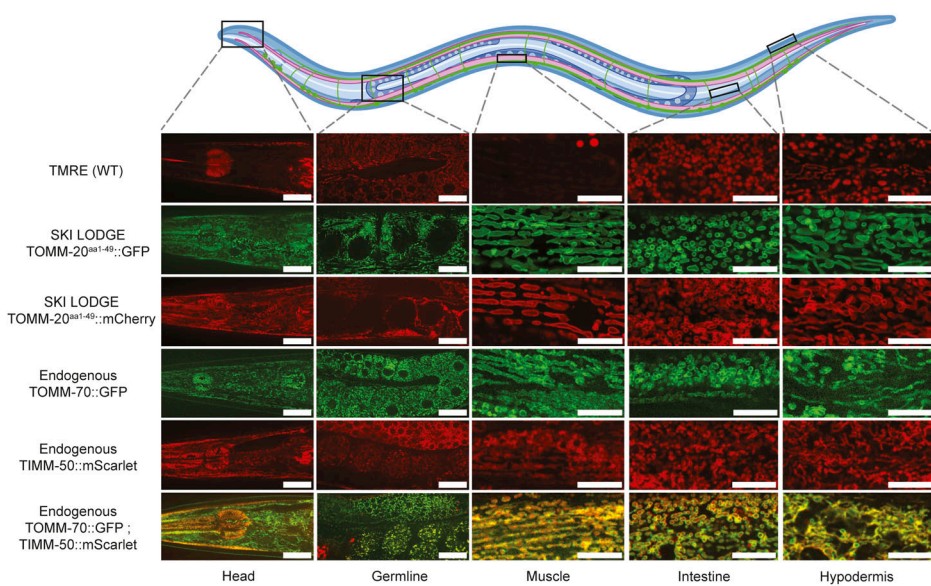

**Figure 2. Improved mitochondrial reporter tools in *C. elegans*.**

Comparative analysis of mitochondrial visualization techniques in various tissues *C. elegans* on day 2 of adulthood. The upper panel of images demonstrates the variable staining efficacy of TMRE in WT N2 worms. The novel SKI LODGE strains, expressing the TOMM-20 transmembrane domain fragment fused to GFP or mCherry, respectively, under the *eft-3p* promoter, exhibit more uniform labeling. Endogenous mitochondrial strains include, GFP expression on the translocase TOMM-70, or mScarlet tagging on the translocase of the inner mitochondrial membrane TIMM-50, respectively. Dual labeling of both mitochondrial membranes is achieved in a strain expressing GFP on the OMM and mScarlet on the IMM. Scale bar (head, germline): 20 $\mu$m. Scale bar (muscle, intestine, hypodermis): 10 $\mu$m.

no silencing (Silva-García et al, 2019). Compared with the mito-GFP strain or staining with dyes, these new lines show more consistent expression and accurate labeling of mitochondrial networks in worms. In these strains, the visualization of mitochondria reaches high intensity and resolution across various tissues including muscle, intestine, and hypodermis (Fig 2), as well as in multiple tissues in the worm's head, such as the pharynx, hypodermal cells, body wall muscle cells, and neuronal cells. Of note, the ectopic *eft-3* promoter in the SKI LODGE system is not expressed in the germline (Fig 2). In addition, the TOMM-20$^{aa1–49}$::GFP/mCherry markers in the SKI LODGE strains do not give any insight into the relative abundance of endogenous mitochondria proteins in the worm. To circumvent these issues, we also used CRISPR to generate a strain with GFP fused to the C-terminus of the endogenous OMM translocase TOMM-70 or, alternatively, mCherry tagged to the translocase of the endogenous IMM, TIMM-50 (Paix et al, 2015) (Fig S2B). Finally, we crossed these strains together to be able to visualize the OMM and IMM together in the same animal, which proves notably useful for studies requiring differentiation between processes exclusive to each of these mitochondrial membranes (Fig 2). Having generated these new tools, we then sought to further characterize their utility and effect on animal physiology.

### Single copy ectopic OMM markers

Using ectopic promoters to drive OMM markers has several advantages. Endogenous mitochondrial membrane proteins remain WT, whereas the higher expression of promoters such as *eft-3* even at single copy increases signal for microscopy. Indeed, a direct comparison of fluorescence intensity between SKI LODGE strains (TOMM-20$^{aa1–49}$::GFP/mCherry) and endogenous strains (TOMM-70::GFP and TIMM-50::mScarlet, respectively) revealed higher intensity levels in the former (Fig 3A and B). This heightened fluorescence intensity facilitates confocal imaging, providing improved clarity and ease of analysis. This difference is evident when comparing tissues between the SKI LODGE and endogenously tagged strains, as fluorophore intensity is noticeably lower in tissues like the body wall muscle and intestine in the TOMM-70::GFP and TIMM-50::mScarlet strains (Fig 3B). Tagging fluorophores to mitochondrial membranes, membrane proteins or proteins involved in mitochondrial remodeling can affect mitochondrial function (Montecinos-Franjola et al, 2020). Reduced mitochondrial function reduces body size, delays development, and decreases reproduction in *C. elegans* (Dillin et al, 2002; Durieux et al, 2011). To determine if the new SKI LODGE ubiquitous TOMM-20$^{aa1–49}$ reporter causes phenotypic differences due to the fluorophore, we therefore measured several healthspan parameters, such as growth, development, and fertility. While the TOMM-20$^{aa1–49}$::GFP animals exhibited consistency in healthspan parameters, including body size, generation time, and brood size, as compared with the WT N2 strain, TOMM-20$^{aa1–49}$::mCherry worms displayed slightly increased generation time and reduced brood size (Fig 3C). This suggests that the introduction of the GFP/mCherry tag to TOMM-20 does not adversely impact these crucial health-related metrics. Lifespan analysis on *Escherichia coli* OP50-1 indicated no significant changes compared with WT animals, although a slight reduction in lifespan was observed on HT115 (Fig 3D). It's important to note that NG plates containing antibiotics were used when using HT115 bacteria, as

our L4440 plasmid confers resistance to carbenicillin. Overexpression of reporters through the SKI LODGE system may therefore exhibit sensitivity to antibiotics, potentially influencing lifespan outcomes.

### Fluorescent tagging of endogenous mitochondrial membrane proteins

Since expression of TOMM-20$^{aa1–49}$::GFP or mCherry in the SKI LODGE strains is driven by an ectopic promoter, it cannot be used to determine relative abundance of endogenous mitochondria proteins in the worm. Having used CRISPR to generate a strain expressing GFP on the C-terminus of endogenous outer membrane translocase TOMM-70 (Paix et al, 2015), we sought further characterize it. This strain facilitates imaging of mitochondrial networks across all tissues, including the germline, which was not possible in previous extrachromosomal strains or the SKI LODGE strain (Fig 2). In line with this, TOMM-70 displays the highest GFP fluorescence intensity in the germline of *C. elegans* and the lowest in the muscle (Fig 4A). The TOMM-70::GFP strain did not exhibit any alterations in body size, generation time or brood size as compared with WT N2 worms (Fig 4B). Similarly, the median lifespan of the TOMM-70::GFP strain was comparable to that of the WT animals. Overall, these observations suggest that the endogenous GFP tag on TOMM-70 does not lead to any overt physiological abnormalities. Contrarily, physiological effects were observed in the TIMM-50::mScarlet animals including altered growth, slowed development and decreased reproductive capacity (Fig 4B). Despite this, the TIMM-50::mScarlet animals had no difference in median lifespan relative to WT animals fed on OP50-1 or HT115 bacteria (Fig 4C). This was also the case for the double TOMM-70::GFP; TIMM-50::mScarlet animals, which showed increased generation time and reduced brood size (Fig 4B and C).

In summary, CRISPR/Cas9 facilitated the generation of new tools to better study mitochondrial networks via fluorescence tagging of mitochondrial outer and inner membrane components, notably with the TOMM-70::GFP strain providing a nuanced view of mitochondrial expression levels across tissues. These tools hold promise for researchers studying mitochondrial morphology changes in live *C. elegans*.

### Modulation of mitochondrial dynamics in *C. elegans* tissues

To test whether our new strains could reveal changes in mitochondrial dynamics, we performed imaging analysis of mitochondria after impairing the expression of the key factors in mitochondrial dynamics, the fission protein DRP-1 and the fusion protein FZO-1 (the homolog of mammalian MFN1/MFN2), respectively. To do this, we assessed mitochondrial morphology in the SKI LODGE TOMM-20$^{aa1–49}$::GFP strain (Fig 5A) and the endogenous TOMM-70::GFP strain (Fig 5B) following RNA interference (RNAi, in OP50 bacteria) targeting DRP-1 and FZO-1. Changes in mitochondrial morphology were observed in all examined *C. elegans* tissues, including the intestine, hypodermis, and body wall muscle (Fig 5A and B). In both strains, DRP-1 RNAi resulted in elongated mitochondrial network morphology, whereas FZO-1 RNAi led to increased mitochondrial fragmentation. Notably, DRP-1 down-regulation also resulted in larger tubules and aggregated mitochondria connected by thin filaments, particularly in the intestine, as previously reported (Weir et al, 2017; Traa & Raamsdonk, 2024 *Preprint*). While mitochondrial remodeling was observed in all tissues, the most

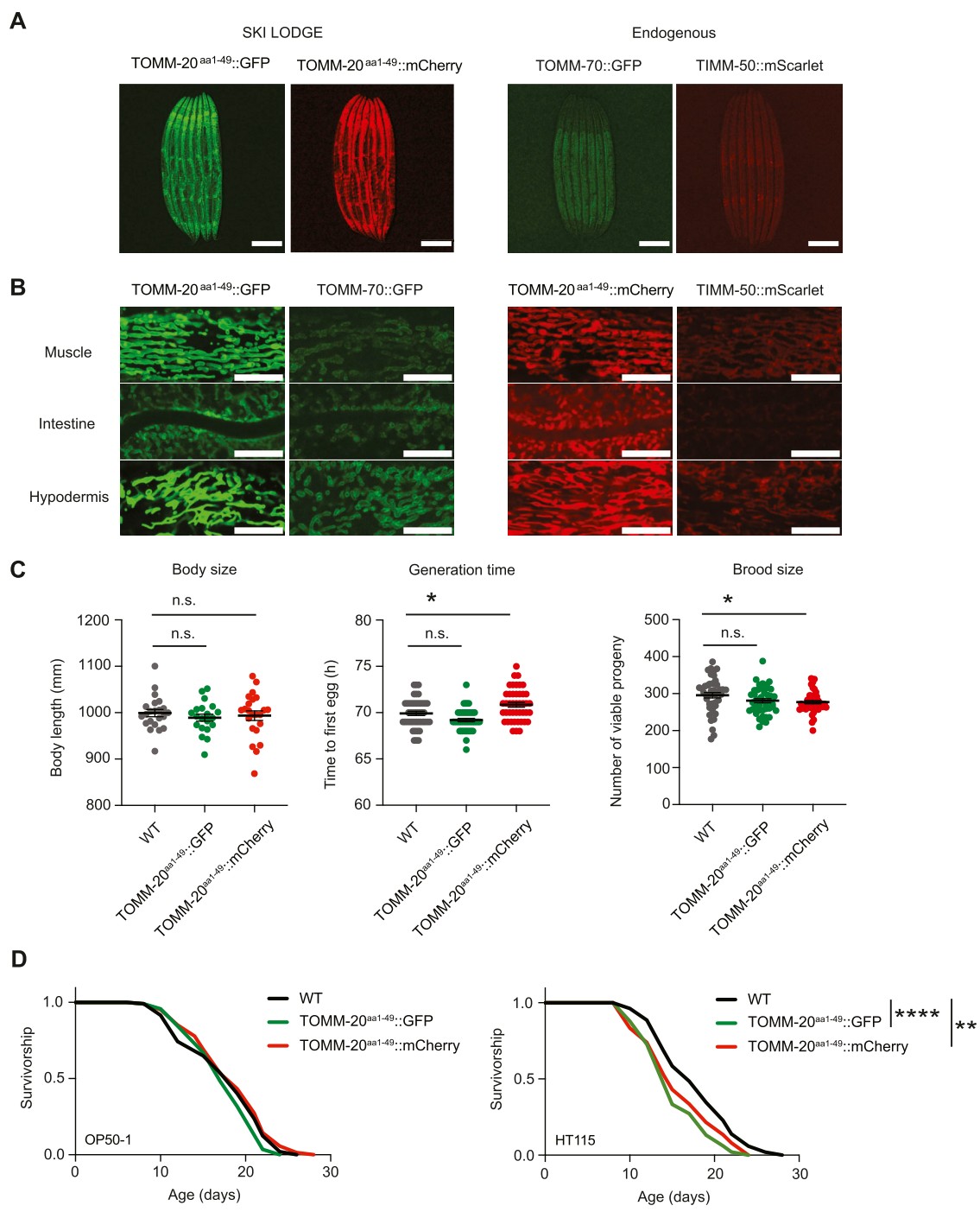

**Figure 3. Novel TOMM-20 reporter strains generated using SKI LODGE in *C. elegans*.**
**(A)** Fluorescence images of the SKI LODGE TOMM-20$^{aa1-49}$::GFP and TOMM-20$^{aa1-49}$::mCherry worms, respectively, as well as the TOMM-70::GFP and TIMM-50::mScarlet worms at day 1 of adulthood. Laser exposure: 65 ms. Scale bar: 200 $\mu$m. **(A, B)** Fluorescence images of the tissues from the strains in (A) at day 1 of adulthood. Laser exposure and brightness/contrast were kept constant for comparison of fluorophore intensity in all tissues. Scale bar: 10 $\mu$m. **(C)** Healthspan analysis of SKI LODGE strains compared with WT N2, showing body size (left), generation time (middle), and brood size (right). Body size measurements were performed with 25 individual worms. Generation time and brood size measurements were performed with 45 worms pooled from three independent experimental repeats. All values are presented as mean ± SEM. *$P < 0.05$ (nonparametric Kruskal–Wallis's test). **(D)** Survival of SKI LODGE strains compared with WT in OP50-1 (left lifespan) and HT115 (right lifespan) bacteria. **$P < 0.01$ indicating significant difference between WT animals and TOMM-20$^{aa1-49}$::mCherry worms, ****$P < 0.0001$ represents significant difference between WT and TOMM-20$^{aa1-49}$::GFP worms by one-way ANOVA with Tukey's multiple comparisons test. See Table S3 for detailed lifespan statistics.

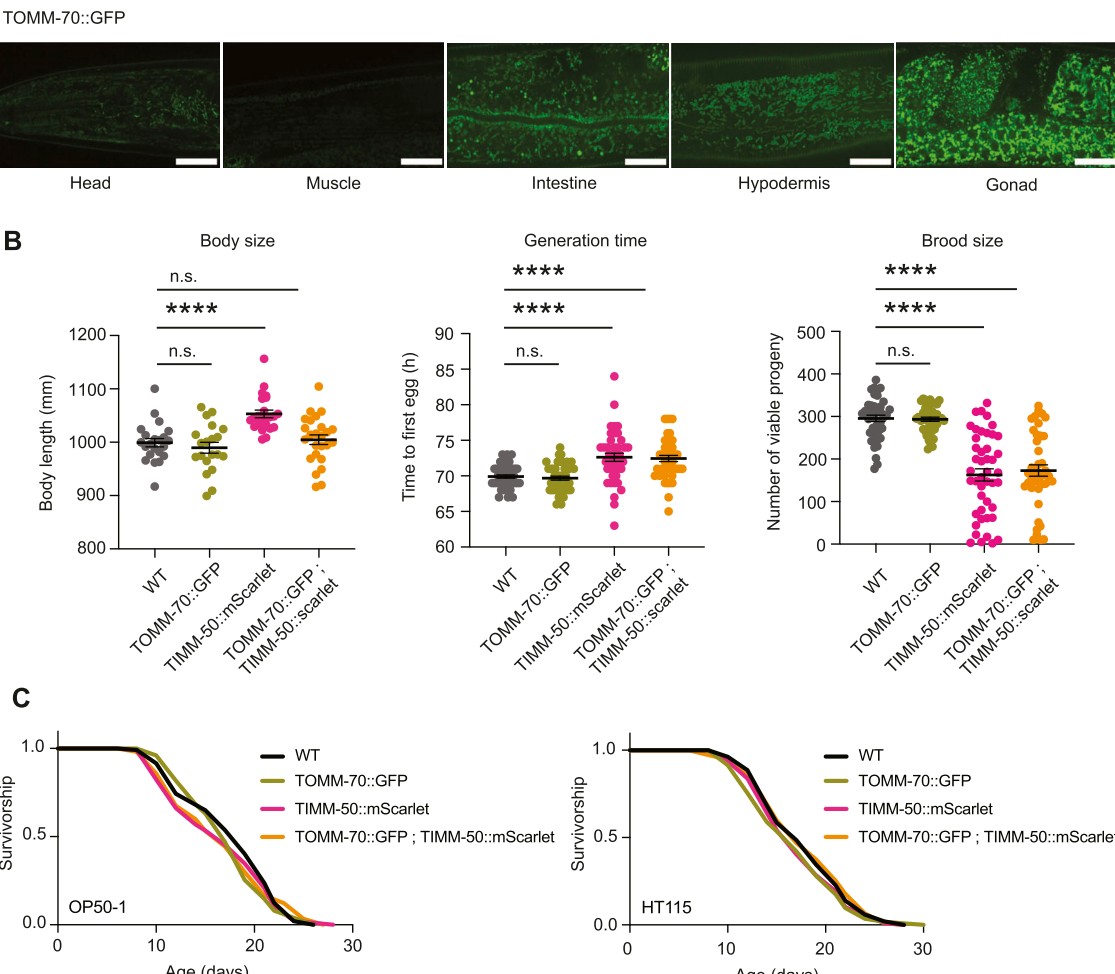

**Figure 4. Endogenous tagging of mitochondrial membrane proteins in *C. elegans*.**
**(A)** Visualization of relative endogenous expression levels of TOMM-70 through the intensity of GFP in various *C. elegans* tissues of the TOMM-70 endogenous strain at day 1 of adulthood. Scale bar: 20 $\mu$m. **(B)** Comparative healthspan analysis among TOMM-70::GFP, TIMM-50::mScarlet, and the double TOMM-70::GFP; TIMM-50::mScarlet strains, respectively, in relation to WT N2 worms. Parameters such as body size (left graph), generation time (middle), and brood size (right) are presented for these strains. Body size measurements were performed with 25 individual worms. Generation time and brood size measurements were performed with 45 worms pooled from three independent experimental repeats. All values are presented as mean ± SEM. ****$P < 0.0001$ (nonparametric Kruskal–Wallis's test) between the indicated groups. **(C)** Survival of endogenous strains compared with WT in OP50-1 (left lifespan) and HT115 (right lifespan) bacteria (analysis by one-way ANOVA with Tukey's multiple comparisons test).

pronounced effects were seen in the muscle as it displays a more balanced baseline of mitochondrial fission/fusion states compared with the higher baseline fragmentation seen in the hypodermis and intestine (Fig 5). Therefore, quantification of mitochondrial dynamics in muscle of animals exposed to DRP-1 RNAi resulted in a shift towards elongated mitochondrial morphology, as reflected by increased mitochondrial length and aspect ratio and decreased circularity. Contrarily, FZO-1 RNAi caused greater mitochondrial fragmentation, indicated by reduced length and aspect ratio and increased circularity measured in the muscle of TOMM-20[aa1–49]::GFP or TOMM-70::GFP animals exposed to FZO-1 RNAi, which is consistent with disrupted fusion.

These observations highlight the potential of these strains for studying mitochondrial dynamics in the multi-tissue system of *C. elegans*. Future research could leverage these strains by crossing them with DRP-1 and FZO-1 knockout mutants to investigate, for example, age-related changes in mitochondrial morphology. In addition, these strains can be used to explore mitochondrial dynamics across short- or long-lived genetic backgrounds, at specific life stages, or under various physiological and stress conditions.

## Discussion

*C. elegans* represent a unique middle ground between single-cell systems and mammals to study organelle biology (Onraet & Zuryn, 2024). No one tool is best equipped to take on this challenge, and instead researchers must decide which tool best serves them depending upon their scientific question, resources, and timeline

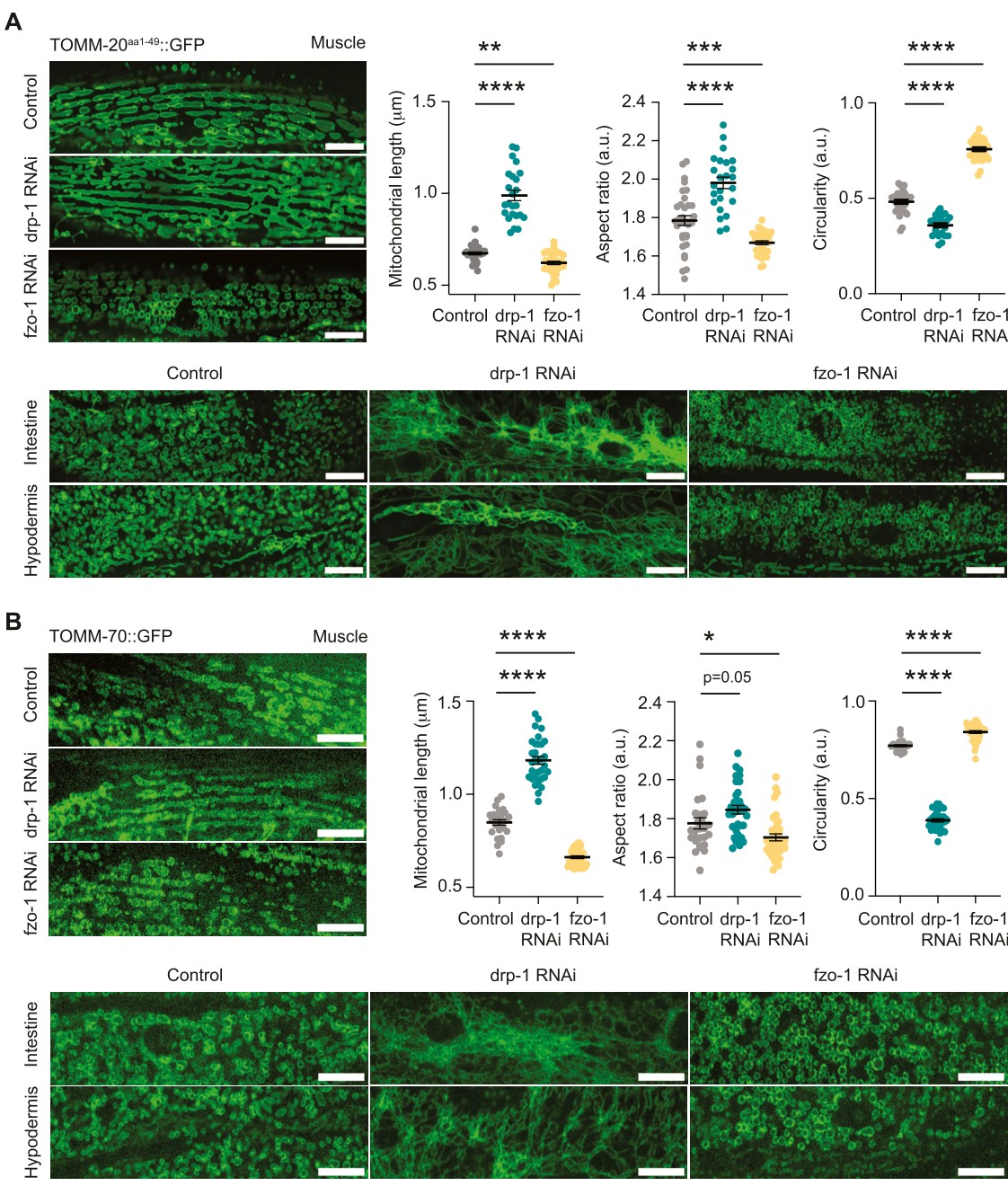

**Figure 5. Analysis of mitochondrial dynamics in *C. elegans* tissues.**
**(A)** Fluorescence images of tissues from SKI LODGE TOMM-20$^{aa1-49}$::GFP worms on day 2 of adulthood after treatment from hatch with control (empty vector), DRP-1, or FZO-1 RNAi in OP50 bacteria, respectively. The images display the body wall muscle along with the corresponding quantification of changes in mitochondrial dynamics, including mitochondrial length, aspect ratio, and circularity, quantified using MitoMAPR. In addition, images of intestine and hypodermis tissues treated with the corresponding RNAis are shown. Scale bar: 10 $\mu$m. For the quantification of mitochondrial shapes, all values are presented as mean ± SEM. **$P < 0.01$, ***$P < 0.001$ ****$P < 0.0001$ indicated versus the control group (nonparametric Kruskal–Wallis's test). **(A, B)** Fluorescence images of tissues from endogenous TOMM-70::GFP worms following the same conditions and analyses as described in (A). Scale bar: 10 $\mu$m. For the quantification of mitochondrial shapes, all values are presented as mean ± SEM. *$P < 0.05$, ****$P < 0.0001$ indicated versus the control group (nonparametric Kruskal–Wallis's test).

(Fig S6). Several methods exist that do not require gene editing, including electron microscopy, immunofluorescence, and cellular dyes such as MitoTracker and TMRE. However, although these all have unique advantages, they all require fixing or termination of the sample, which limits their utility for longitudinal and noninvasive live imaging. The introduction of novel *C. elegans* strains with fluorophore-tagged components of the outer and inner mitochondrial membrane marks a significant advancement in our ability to investigate mitochondrial dynamics within intact living organisms. Here, we introduce novel fluorophore-based genetic tools for imaging mitochondrial networks in *C. elegans*, addressing the limitations of existing tools (Table S1).

Using targeted gene-editing strategies to generate fluorescent reporters of mitochondrial membranes offers substantial advantages over extrachromosomal methods as it allows stable and uniform expression within an isogenic population of animals. This is critical for accurate quantitative analysis of mitochondrial dynamics. It also improves upon gamma irradiation methods of integrating edits into the *C. elegans* genome as it has less off-target effects, and bypasses the need for co-selection. Our SKI LODGE strains have increased fluorophore intensity, which is homogeneous across different tissues and allows straightforward imaging using basic microscopes without much need for imaging processing. For analysis of endogenous protein expression levels, strains such as the TOMM-70::GFP or TIMM-50::mScarlet can be used, albeit with reduced fluorophore intensity and the need for fine-tuning of higher resolution image acquisition.

How the tool impacts the function of mitochondria themselves or organismal physiology is also a critical consideration, especially when conducting longevity studies and imaging late in life. Whereas the mito-GFP line showed significant reduction in body size, brood size, and delayed development, neither the SKI LODGE lines show overt alterations in healthspan indicators. Although the mean generation time of the SKI LODGE TOMM-20$^{aa1-49}$::mCherry strain (70.84 h ± 0.3266) was significantly different as compared with WT (69.91 h ± 0.3474) animals, this effect is negligible. Similarly, this line shows mildly reduced brood size. Moreover, both SKI LODGE strains display a slight lifespan reduction when aged on NG plates with antibiotics fed on HT115. Maintenance of *C. elegans* in the laboratory standardly includes the removal of spontaneously occurring microbial contaminants to limit experimental variations, since diet, including the presence of contaminants, has been shown to exert potent influences over animal physiology, such as development, reproduction, healthspan, and longevity (Stuhr & Curran, 2020). This is routinely achieved by use of a single or combination of antibiotics. Bacterial food sources have also been engineered to carry specific resistance for selection, and this is particularly relevant for RNA interference studies in *C. elegans*, where worms are fed RNAi-containing bacteria. In these studies, carbenicillin is consistently included to guarantee the exclusive presence of RNAi or control bacteria carrying resistance to carbenicillin (Conte et al, 2015). However, many antibiotics widely used in research have profound effects on mitochondrial function and dynamics, which ultimately affect cell viability, metabolism, and organismal physiology (Houtkooper et al, 2013; Moullan et al, 2015). Indeed, we observed mildly increased mitochondrial fragmentation in worm tissues exposed to carbenicillin and/or kanamycin (Fig S3). Altogether, these observations suggest that antibiotics might induce mitochondrial fragmentation in *C. elegans*. Therefore, there is a clear need to define cleaner research tools that do not affect the function of mitochondria, as well as greater awareness of the undesired cofounders caused by antibiotics.

Which fluorophore is used can also impact the usefulness of the tool. We observed the development of protein aggregates linked to aging in strains utilizing red fluorophores, irrespective of whether they were tagged to mScarlet, RFP, or mCherry (Fig S4). This occurrence is unique to red fluorophore tags and does not manifest in GFP strains (Fig S4). Moreover, this observations are not contingent on the specific protein to which these fluorophores are attached, as evidenced by the formation of protein aggregates in both the TOMM-

20$^{1-49}$::mCherry strain and the TIMM-50::mScarlet (Fig S4). Consequently, in longitudinal studies involving the aging of mitochondrial reporter strains, it is advisable to refrain from using these red fluorophores. This phenomenon may provide an explanation for the adverse impacts on growth and reproductive capacity observed in the endogenous TIMM-50::mScarlet strain, effects not seen in TOMM-70 animals with an endogenous GFP tag (Fig S4). Previous research has shown the formation of granular aggregates in immortalized hAECs transfected with mitochondria-targeted mCherry and DSRED (Taiko et al, 2022). These aggregates were not due to increased mitochondrial fragmentation or mitophagy, as evidenced by their lack of staining with MitoTracker (Taiko et al, 2022). Some cells showed coexistence of both aggregated and properly targeted mCherry and DSRED. This aligns with our findings in *C. elegans*, where the aggregates of fluorophores exhibit increased fluorescence intensity, whereas the mitochondria show subtle staining.

There is vast potential for advancing tools to investigate organelle biology and inter-organelle communication within mitochondrial research. In this study, we have introduced several novel imaging tools for effectively visualizing mitochondria in *C. elegans*. Nevertheless, we acknowledge the existence of others. For instance, a COX-4::GFP fusion protein strain is commonly used for visualizing mitochondria in *C. elegans*, where COX-4 is tagged at the C-terminus with eGFP using genome editing methods (Raiders et al, 2018). When comparing the fluorescence intensity pattern of the COX-4::GFP strain with our TOMM-70::GFP strain or the SKI LODGE TOMM-20$^{aa1-49}$::GFP strain, the COX-4::GFP animals exhibit an intensity level intermediate between the other two (Fig S5). While we highlighted the advantages of our TOMM-70::GFP strain for visualizing mitochondria in the germline, other strains are also available. For example, SJZ106, which expresses TOMM-20::mKate2::HA specifically in the germline, can be used for fluorescent imaging of germline mitochondrial morphology and tissue-specific mitochondrial isolation (Ahier et al, 2018). In fact, the Zuryn laboratory has introduced a variety of *C. elegans* strains designed for the tissue-specific isolation of mitochondria (Ahier et al, 2018). In addition, there are several endogenously tagged mitochondrial protein strains, such as ACO-2::GFP, which has been characterized for visualizing prototypical changes in mitochondrial morphology associated with aging and cellular stress (Begelman et al, 2022).

To address the challenge of inserting large fluorophore-sized DNA segments into the *C. elegans* genome, the split-wrmScarlet technique offers a more efficient method for fluorescently labeling endogenous *C. elegans* proteins (Goudeau et al, 2021). This method enables single-color labeling in germline or muscle cells and dual-color labeling in somatic cells. In addition to the common use of widely recognized fluorophores like GFP or RFP, there are emerging alternatives. These new fluorophores not only serve the purpose of staining mitochondria but also contribute to evaluating various parameters related to mitochondrial function. This is the case of Supernova, a monomeric variant of KillerRed which allows for the evaluation of ROS levels by chromophore-assisted light inactivation in many living specimens including *C. elegans* (CALI) (Takemoto et al, 2013; Onukwufor et al, 2022). In addition to this, mt-Keima consists of a pH-dependent fluorescence assay based on the coral-derived protein Keima targeted to the mitochondrial matrix (Sun et al,

2015). In this case, mt-Keima allows for the detection of mitophagy by a shift in the excitation wavelength from the physiological pH of the mitochondria (pH 8.0) to the lysosome (pH 4.5) upon engulfment of the mitochondria by the autophagosome (Sun et al, 2015, 2017). While not yet implemented in *C. elegans*, this method has demonstrated success in cultured cells (Sun et al, 2017) and unfixed mouse tissues (Katayama et al, 2011; Sun et al, 2015, 2017).

With the relative ease of CRISPR insertion in *C. elegans*, and safe harbor knock-in systems such as SKI LODGE (Silva-García et al, 2019), the future is bright for expanding this model system for studying mitochondria, additional organelles, and their interactions. Our comparative analysis highlights the importance of selecting appropriate tools based on specific experimental goals (Table S1 and Fig S6). The significance of these newly generated tools extends beyond this study, as we anticipate their widespread use within the scientific community working on *C. elegans*. We envision that the application of these tools will open avenues for further exploration and discoveries in the realm of mitochondrial research.

## Materials and Methods

### *C. elegans* strains and husbandry

Worms were grown and maintained on standard nematode growth media (NGM) seeded with *E. coli* (OP50-1), and maintained at 20°C. OP50-1 was cultured overnight in LB at 37°C, after which 100 $\mu$l of liquid culture was seeded on plates to grow for 2 d at room temperature. HT115 was cultured overnight in LB containing carbenicillin (100 $\mu$g/ml) and tetracycline (12.5 $\mu$g/ml). Then, 100 $\mu$l of liquid culture was seeded on NGM plates containing 100 $\mu$g/ml carbenicillin. To make WBM1231 (SKI LODGE TOMM-20$^{aa1-49}$::GFP) and WBM1232 (SKI LODGE TOMM-20$^{aa1-49}$::mCherry), CRISPR/Cas9 was used to insert *tomm-20aa1–49::GFP* or *tomm-20aa1–49::mCherry*, respectively, into WBM1140 (Silva-García et al, 2019). To make WBM1444 (TOMM-70::GFP), CRISPR/Cas9 was used to insert GFP on the carboxyl terminus of *tomm-70* at its endogenous locus in the *C. elegans* genome. Similarly, mScarlet was inserted on the C-terminus of the *scpl-4* gene (TIMM-50) at the endogenous *C. elegans* locus to generate WBM1688 (TIMM-50::wrmScarlet). To generate the *myo-3*::*gfp^{mt}(zcIs14)* (SJ4103) strain, which we obtained from CGC, the *myo-3*::*GFP^{mit}* plasmid expressing a mitochondrially localized GFP (GFPmt) with a cleavable mitochondrial import signal peptide (Labrousse et al, 1999; Benedetti et al, 2006) was integrated stably according to previous instructions. N2 Bristol WT and COX-4::GFP (JJ2586) were obtained from CGC. All strains are described in Table S2.

### CRISPR/Cas9 gene editing

All gene edits were performed based on previously described protocols (Paix et al, 2015; Silva-García et al, 2019). Homology repair templates were amplified by PCR using primers that introduced a minimum of 35 base pairs of homology flanking the site of insertion at both ends. CRISPR injection mixes were generated with the following composition: 2.5 $\mu$l tracrRNA (4 $\mu$g/$\mu$l), 0.6 $\mu$l dpy-10 crRNA (2.6 $\mu$g/$\mu$l), 0.5 $\mu$l target gene crRNA (2.6 $\mu$g/$\mu$l), 0.25 $\mu$l dpy-10 ssODN

(500 ng/$\mu$l), homology repair template (200 ng/$\mu$l final in the mix), 0.375 $\mu$l Hepes pH 7.4 (200 mM), 0.25 $\mu$l KCl (1 M) and RNase free water to make up the volume to 8 $\mu$l. Prior to injection, 2 $\mu$l purified Cas9 (12 $\mu$g/$\mu$l) was added before the solution was centrifuged at 13,000 rpm and then incubated at 37°C for 10 min (Eppendorf 5424 Microcentrifuge). Mixes were microinjected into the germline of day 1 adult hermaphrodites as previously described (Mello et al, 1991; Silva-García et al, 2019). Worms generated using CRISPR were outcrossed at least six times before being used for experiments to remove the co-injection marker phenotype and other off-target edits.

### Lifespans

Lifespans assays were performed at 20°C on 6 cm NGM plates seeded with OP50-1, or seeded with HT115 containing 100 $\mu$g/ml carbenicillin. All worms were kept fed for at least three generations after thawing to minimize transgenerational effects of starvation on lifespan (Rechavi et al, 2014). Then, worms were synchronized through timed egg lays using day 1 gravid adults. When progeny from the egg lay reached day 1 of adulthood, 120 worms were transferred to fresh plates at 20 worms per plate. When excessive censoring was anticipated, the initial population consisted of 160 worms. For the lifespan assays involving HT115 bacteria, worms underwent prior bleaching and were left for two generations before initiating the egg lay. When working with genotypes that were developmentally delayed, bleaches and egg lays were staggered so that all worms reached day 1 adult stage simultaneously with the WT animals. Worms were transferred to freshly seeded plates every day until reaching day 4 to separate from progeny, and then every other day until day 10 or 11 of adulthood. Survival was assessed every other day, with worms considered dead when unresponsive to three touches on the head and tail. Worms were censored if they crawled up the wall, bagged, or exploded. Additional details and statistical information can be found in the Table S3.

### Generation time assay

Animals were synchronized by performing an egg lay 3 d prior to the experiment. The egg lay for TIMM-50 and SJ4103 animals occurred 3 h earlier than the other strains due to their developmental delay, whereas the egg lay for all other strains took place in the evening. During the egg lays, 20 animals were placed on 2 d seeded NGM plates, allowed to lay eggs for 30 min, and then removed. Early in the morning 3 d later, 15 L4 animals for each condition were selected and placed onto individual 3.5 cm NG plates, which had been seeded 1 d prior with 50 $\mu$l of OP50-1 bacteria. Animals were monitored and scored every hour until the first egg was laid.

### Brood size measurement

Animals were synchronized by egg lay onto NGM plates with OP50-1 bacteria following the instructions in "Generation time assay." At the L4 stage, 15 animals from each condition were placed individually on 3.5 cm NG plates seeded with OP50-1 bacteria. During the consecutive 2 d, animals were transferred to freshly seeded 3.5 cm NG plates twice per day, once in the morning and then in the

evening. During the third and fourth days of the assay, animals were transferred only once. The plates containing the eggs were kept in the incubator at 20°C for 3 d until the progeny developed into day 1 adults. The number of progeny on each plate was documented, and the daily counts were summed for each parent individually.

### Body size measurement

Worms at day 1 of adulthood were anesthetized on NGM plates without bacteria using 1 mg/ml tetramisole. Once still, worms were imaged on a Zeiss Discovery V8 dissection microscope with an Axiocam camera. All animals were imaged in brightfield at 5x with a constant exposure of 65 ms. Body length was analyzed in ImageJ by drawing a line end-to-end down the midline of the worm and measuring the length of the line.

### Confocal microscopy

Worms were anesthetized in 0.1 mg/ml tetramisole in 1X M9 buffer on empty NGM plates and mounted on thin 2% agarose pads on glass slides with 0.05 mm Polybead microspheres (Polysciences) for immobilization. A No. 1.5 cover glass was gently placed on top of worms and sealed with clear nail polish. Images were performed on a Yokogawa CSU-X1 spinning disk confocal system (Andor Technology) with a Nikon Ti-E inverted microscope (Nikon Instruments), using a Plan-Apochromat 100x/1.45 objective lens. Images were acquired using a Zyla cMOS camera and NIS elements software was used for acquisition parameters, shutters, filter positions and focus control. Images were taken from at least 10 worms per condition, with a minimum of two independent experiments performed. For the imaging of the endogenous strains where the fluorophore intensity is dimmer, a binning of 2 × 2 was used. For staining with TMRE (T669; Thermo Fisher Scientific), WT N2 worms were placed on OP50-1 seeded NGM plates with 10 $\mu$M TMRE for 24 h prior to imaging. To analyze changes in mitochondrial dynamics, worms were exposed to *drp-1* or *fzo-1* RNAi grown on OP50 bacteria. Expression of dsRNA was induced by adding 100 $\mu$l of 100 mM isopropyl $\beta$-d-1-thiogalactopyranoside (IPTG) solution onto OP50 lawns before placing the worms. Worms were synchronized by egg lay from gravid adults. Once the progeny reached day 2 of adulthood, 30–35 worms were imaged to assess changes in mitochondrial morphology.

### Statistical analysis

Data were graphed and analyzed in GraphPad Prism 9. For generation time, brood size, body size, data were evaluated for normality and analyzed by the nonparametric Kruskal–Wallis's test. For lifespan experiments, survival curves were analyzed using the Log-rank (Mantel–Cox) test. A one-way ANOVA followed by a multiple comparisons test was performed to identify statistically significant differences between groups. Quantification of mitochondrial dynamics changes was performed using MitoMAPR (Zhang et al, 2019). Circularity and aspect ratio (AR) were analyzed according to the indications in Valera-Alberni et al (2021). Mitochondria exhibiting a perfect circular shape have a circularity value close to 1.0, whereas more elongated mitochondria have a circularity value that is closer to 0.0. AR is calculated as ([major axis]/[minor axis]) and reflects the "length-to-width ratio." An AR value of 1 indicates a perfect circle and its value increases as mitochondria elongate. For all figures: * indicates $P < 0.05$, ** indicates $P < 0.01$, *** indicates $P < 0.001$, **** indicates $P < 0.0001$.

## Supplementary Information

## Acknowledgements

Funding support was provided by NIH/NIA R01AG044346, R01AG067106 and R21AG056930. P Yao was funded by F31AG066458. Fig S6 was created with BioRender. We thank the *Caenorhabditis* Genetic Center for providing strains. We also thank the Mair laboratory members for comments and discussion on the project and manuscript.

### Author Contributions

M Valera-Alberni: conceptualization, resources, data curation, formal analysis, investigation, methodology, and writing—original draft, review, and editing.
P Yao: conceptualization, resources, investigation, methodology, and writing—original draft.
S Romero-Sanz: data curation, formal analysis, investigation, and methodology.
A Lanjuin: resources, data curation, formal analysis, investigation, and methodology.
WB Mair: conceptualization, resources, funding acquisition, validation, methodology, project administration, and writing—original draft, review, and editing.

### Conflict of Interest Statement

The authors declare that they have no conflict of interest.

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
