## [Reviewer comments · Life Science Alliance]

Novel Imaging Tools to Study Mitochondrial Morphology in *Caenorhabditis elegans*

Miriam Valera-Alberni, Pallas Yao, Silvia Romero-Sanz, Anne Lanjuin and William B Mair

DOI: 10.26508/lsa.202402918

Corresponding author(s): Dr. William B Mair (Harvard University)

Review timeline:

Submission Date:	2024-07-01
Editorial Decision:	2024-07-02
Revision Received:	2024-08-23
Editorial Decision:	2024-08-26
Revision Received:	2024-08-28
Accepted:	2024-08-29

Scientific Editor: Eric Sawey

Transaction Report:

Please note that the manuscript was previously reviewed at another journal and the reports were taken into account in the decision-making process at Life Science Alliance.

No Peer Review Process File is available with this article, as the authors have chosen not to make the review process public in this case.

Re: Life Science Alliance manuscript #LSA-2024-02918-T

William B Mair

Harvard School of Public Health

677 Huntington Ave

Boston, Massachusetts 02115

Dear Dr. Mair,

Thank you for submitting your manuscript entitled "Novel Imaging Tools to Study Mitochondrial Dynamics in *Caenorhabditis elegans*" to Life Science Alliance. We invite you to submit a revised manuscript addressing the Reviewer comments. Reviewer 1's requests to study mitochondrial dynamics in various settings can either be addressed, or the title and associated claims toned down. The Reviewer's figure-specific comments should be addressed.

Thank you for this interesting contribution to Life Science Alliance. We are looking forward to receiving your revised manuscript.

Sincerely,

Eric Sawey, PhD

Executive Editor

Life Science Alliance

<http://www.lsjournal.org>

B. MANUSCRIPT ORGANIZATION AND FORMATTING:

RE: Life Science Alliance Manuscript #LSA-2024-02918-TR

Dr. William B Mair
Harvard University
Harvard school of public health
677 Huntington ave
boston, Massachusetts 02115

Dear Dr. Mair,

Thank you for submitting your revised manuscript entitled "Novel Imaging Tools to Study Mitochondrial Morphology in *Caenorhabditis elegans*". We would be happy to publish your paper in Life Science Alliance pending final revisions necessary to meet our formatting guidelines.

- please be sure that the authorship listing and order is correct
- please upload your manuscript text file as an editable doc file
- please upload any table files as editable doc files
- please add the supplementary figure legends to the main manuscript file
- please add a Running Title for your manuscript to our system
- please add an alternate abstract/summary blurb to our system
- please add the Twitter handle of your host institute/organization as well as your own or/and one of the authors in our system
- In your manuscript, you sometimes refer to the supplementary figures as EV figures in your figure callouts. Please update this and refer to these figures as supplementary figures.

LSA now encourages authors to provide a 30-60 second video where the study is briefly explained. We will use these videos on social media to promote the published paper and the presenting author (for examples, see <https://docs.google.com/document/d/1-UWCfbE4pGcDdcgzcmiuJI2XMBJnxKYeqRvLLrLS08s/edit?usp=sharing>). Corresponding or first-authors are welcome to submit the video. Please submit only one video per manuscript. The video can be emailed to contact@life-science-alliance.org

A. FINAL FILES:

B. MANUSCRIPT ORGANIZATION AND FORMATTING:

Thank you for your attention to these final processing requirements. Please revise and format the manuscript and upload materials within 5 days.

Sincerely,

RE: Life Science Alliance Manuscript #LSA-2024-02918-TRR

Dr. William B Mair
Harvard University
Harvard school of public health
677 Huntington ave
boston, Massachusetts 02115

Dear Dr. Mair,

Thank you for submitting your Methods entitled "Novel Imaging Tools to Study Mitochondrial Morphology in *Caenorhabditis elegans*". It is a pleasure to let you know that your manuscript is now accepted for publication in Life Science Alliance. Congratulations on this interesting work.

DISTRIBUTION OF MATERIALS:

Again, congratulations on a very nice paper. I hope you found the review process to be constructive and are pleased with how the manuscript was handled editorially. We look forward to future exciting submissions from your lab.

Sincerely,
